# Affordance-Driven Next-Best-View Planning for Robotic Grasping

**Xuechao Zhang[1,2,*], Dong Wang[2,✉], Sun Han[1,2], Weichuang Li[2],**
**Bin Zhao[2], Zhigang Wang[2], Xiaoming Duan[1], Chongrong Fang[1], Xuelong Li[2], Jianping He[1,✉]**

[1]Shanghai Jiao Tong University, [2]Shanghai Artificial Intelligence Laboratory

**Abstract:** Grasping occluded objects in cluttered environments is an essential component in complex robotic manipulation tasks. In this paper, we introduce an AffordanCE-driven Next-Best-View planning policy (ACE-NBV) that tries to find a feasible grasp for target object via continuously observing scenes from new viewpoints. This policy is motivated by the observation that the grasp affordances of an occluded object can be better-measured under the view when the view-direction are the same as the grasp view. Specifically, our method leverages the paradigm of novel view imagery to predict the grasps affordances under previously unobserved view, and select next observation view based on the highest imagined grasp quality of the target object. The experimental results in simulation and on a real robot demonstrate the effectiveness of the proposed affordance-driven next-best-view planning policy. Project page: https://sszxc.net/ace-nbv/.

**Keywords:** Grasp Synthesis, Neural SDF, Next-Best-View Planning

## 1 Introduction

When we aim to pick up an occluded object from an unstructured environment, observations from a single perspective often fail to provide sufficient affordances information, and we spontaneously move our heads to obtain new perspectives of the occluded object. The driving force behind the actions that lead us to seek the next observation relies on our imagination and spatial reasoning abilities. We know in which direction we can better interact with objects, and we will choose to observe in this direction the next time. However, current intelligent robotic systems are not able to perform these tasks efficiently, and a unified framework for addressing this challenge is lacking. In this work, we aim to investigate the feasibility of endowing robots with this capability.

As shown in Fig. 1, we focus on the task of grasping a specific object in cluttered scenes by a robotic arm with a parallel-jaw gripper. There are relatively mature approaches for predicting grasp poses for unknown objects in cluttered scenes, and most of them first observe the entire scene from one or more fixed viewpoints [1, 2] and predict grasps of all objects at once. However, these methods may fail to predict a feasible grasp for a specific object due to heavy occlusions between objects. In order to design a more stable grasp affordances prediction pipeline, some previous works [3, 4] introduced active perception modules to observe the scene from several new selected viewpoints before executing the grasp. They all select the new observation view directions based on the information gain of object geometry reconstruction. However, the improvement of geometry reconstruction does not always indicate a better grasp quality.

In this paper, as shown in Fig. 1, we build on the intuition that the grasp affordances can be better-measured using the observation when the observation view direction is the same as the grasp direction. Based on this insight, we leverage the novel view imagery ability of the implicit neural

---

* Work done while the author was an intern at Shanghai Artificial Intelligence Laboratory.

✉ Corresponding authors: Dong Wang (wangdong@pjlab.org.cn), Jianping He (jphe@sjtu.edu.cn)

7th Conference on Robot Learning (CoRL 2023), Atlanta, USA.

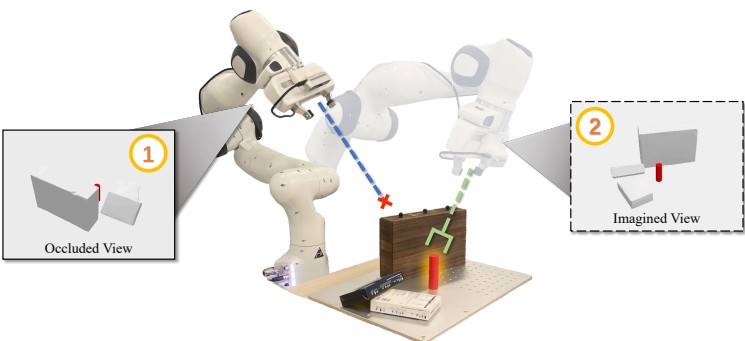

Figure 1: The task of the robotic arm is to grasp the red target object, but the view of its in-hand camera is hindered by a cluttered scene, making it difficult to provide high-quality and collision-free grasping pose predictions. In this study, we draw insight that the grasp affordance of occluded target object can be well-measured when the view direction is the same as the grasp direction, and propose a framework that plans the next observation based on the increment of grasp affordances to find a feasible grasp on the target object.

representation to predict the grasp affordances of imagined novel grasps, and set the next best observation view to the imagined grasp view that yields the highest gain in the grasp quality rather than object geometry reconstruction. Specifically, we first propose a view-aware grasp affordances prediction module to effectively exploit the target object geometry information and occlusion relationships between objects for better grasp synthesis. Then, we adopt similar training paradigm as NeRF [5] to enable our model to imagine the scene representation from previously unobserved viewpoints. With this scene imagery ability, we predict the grasp affordances of target object under many imagined views with proposed view-aware grasp affordances prediction module. Last, a next-best-view planning policy is designed to continuously observe the scene from selected new views until a feasible grasp on target object is found. In summary, the contributions of this work are as follows:

- We propose a view-aware grasp affordances prediction module for better grasp synthesis on an occluded target object in cluttered environments.

- We design a next-best-view planning framework that leverages the implicit neural representation to jointly predict imagined grasp affordances under unseen views and select the next observation view based on the grasp affordances prediction.

- We demonstrate significant improvements of our model over the state-of-the-art for the grasp task in cluttered scenes in simulation and on a real robot.

## 2 Related Works

### 2.1 Grasp Detection

Grasping objects is one of the fundamental abilities for robotic manipulators in manipulation tasks. In order to grasp diverse unknown objects in any environment, a robotic system must effectively utilize the geometric information gathered from its sensors to calculate the feasible grasping poses. Recent advances in deep learning methods have led to rapid developments in robot object grasping [6, 7, 8, 9, 10, 11, 12]. A significant portion of these methods do not require object localization and object pose estimation, but instead perform grasp affordances prediction using end-to-end approaches [1, 13]. In particular, Dex-Net [13, 14] adopts a two-step generate-and-evaluate approach for top-down antipodal grasping, and VGN [1] introduced a one-step approach for predicting 6-DoF grasping configurations in cluttered environments. GIGA [2] exploits the synergistic relationships between the grasp affordances prediction and 3D reconstruction of scenes in cluttered environments for grasp detection. These works all take fixed single or multiple images as input [1, 2, 13, 15], and the robustness of these methods is largely influenced by the observation camera viewpoints [16]. When dealing with complex environments with strong occlusions, many works [3, 4, 17] try to grasp objects by dynamically moving the observation sensors to obtain additional scenes and object geometry information.

## 2.2 Next-Best-View Planning

Next-Best-View (NBV) planning, which aims to recursively plan the next observation position for sensors, is one of the most challenging problems in active vision for robotics [18]. Compared to the passive observation paradigm, active perception with next-best-view planning enables a more flexible way of obtaining environment information. It has been applied in various fields, such as object reconstruction [19, 20, 21], object recognition [22, 23], and grasp detection [3, 4, 17]. NBV planning is typically divided into two categories: synthesis methods and search methods. Synthesis methods directly calculate the next observation position based on current observations and task constraints [24], with some methods [25] working within the paradigm of reinforcement learning. On the other hand, search methods first generate a certain number of candidate viewpoints and then select viewpoints based on human-designed criteria. Most search-based approaches use the gain of 3D geometry reconstruction as the metric to select next viewpoints [3, 4, 19]. In particular, Arruda et al. [3] propose a next-best-view planning policy that maximizes object surface reconstruction quality between the object and a given grasp. Breyer et al. [4] design a closed-loop next-best-view planner based volumetric reconstruction of the target object. Recent work based on neural radiance fields has also proposed some uncertainty-driven methods [26, 27, 28, 29, 30]. However, for grasp tasks, evaluating viewpoints from grasp affordances is a more direct approach [17]. In this article, we mainly explore how to use grasp affordances information for NBV planning.

## 3 Problem Formulation

We consider the same active grasp problem as in [4]: picking up an occluded target object in cluttered scenes using a robotic arm with an eye-in-hand depth camera. As shown in Fig. 2, the target object is partly visible within the initial camera view field and a 3D bounding box is given to locate the target object. Our goal is to design a policy that moves the robotic arm to find a feasible grasp for the target object.

An overview of the whole system is shown in Fig. 2. Given a cluttered scene on a tabletop and an occluded target object $\mathbf{T}$ with corresponding bounding box $\mathbf{T}_{\text{bbox}}$, we aim to predict a feasi-

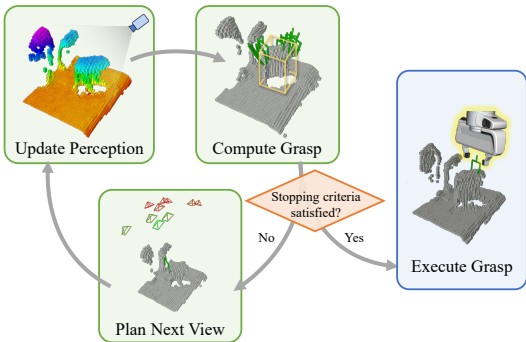

Figure 2: Overview of the next-best-view planning policy for grasping.

ble 6-DoF grasp $\mathbf{G}$ for the target object $\mathbf{T}$. Specifically, at each time $t$, we obtain the observation $\mathbf{D}_t$ and integrate it into a Truncated Signed Distance Function (TSDF) $\mathbf{M}_t$. Then we predict several possible grasps $\mathbf{G}_1, \mathbf{G}_2, \ldots, \mathbf{G}_N$ for the target object based on current $\mathbf{M}_t$. Next, we use a stopping criterion to determine whether a feasible grasp on the target object has been found. If the stopping criterion is satisfied, we select the grasp of $\mathbf{G}^*$ with the highest predicted quality to execute. Otherwise, our proposed model computes a Next-Best-View $\mathbf{O}_{t+1}$ and moves the robotic arm to this viewpoint to get a new observation $\mathbf{D}_{t+1}$ which will be integrated into $\mathbf{M}_{t+1}$. Then, a set of new grasps are predicted using $\mathbf{M}_{t+1}$. This observe-predict-plan closed-loop policy is continuously running until the stopping criterion is met.

## 4 Method

We now present the AffordanCE-driven Next-Best-View planning policy (ACE-NBV), a learning framework that leverages the paradigm of novel view imagery to predict the grasp affordances for the unseen views and achieve the closed-loop next-best-view planning according to predicted grasp affordances. As shown in Fig. 3, our model is composed of two modules: 1) a view-aware grasp affordances prediction module, and 2) an affordance imagery of unseen novel views module. We discuss these two modules in Sec. 4.1 and Sec. 4.2, respectively, followed by the proposed next-best-view planning policy in Sec. 4.3 and training details in Sec. 4.4.

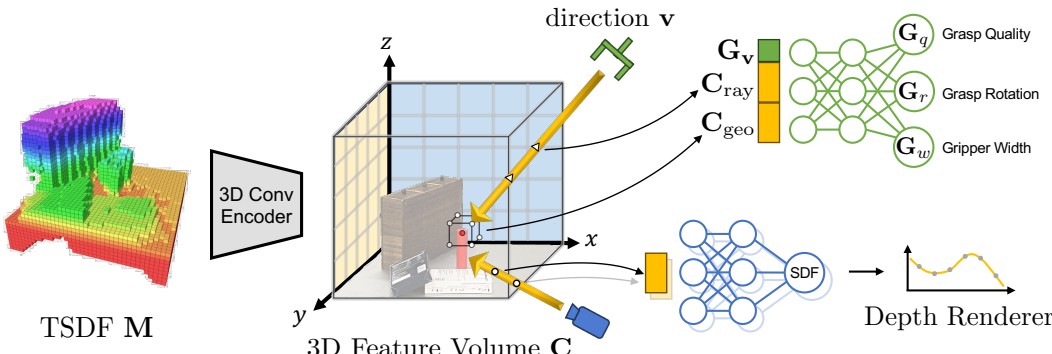

Figure 3: Architecture of the proposed ACE-NBV. The input is a TSDF voxel field $\mathbf{M}$ obtained from the depth image. The upper branch predicts the grasp affordances for the target object and the lower branch synthesizes the depth image of different views, including the previously unseen views. Both branches share the same tri-plane feature volume $\mathbf{C}$.

## 4.1 View-Aware Grasp Affordances Prediction

In this work, we define the grasp affordances in the form of grasp quality $\mathbf{G}_q$, grasp center $\mathbf{G_p} = (x, y, z)$, grasp orientation $\mathbf{G_R} \in SO(3)$, and opening width $\mathbf{G}_w$ of the parallel-jaw gripper. Following the grasp pose representation in [31], we decouple the grasp orientation $\mathbf{G_R}$ as the grasp view $\mathbf{G_v}$ and in-plane rotation $\mathbf{G}_r$. Because of the heavy occlusions in the cluttered scenes, we draw insight that the grasp affordances can be better estimated using the observation whose the view-direction $\mathbf{O_v}$ is the same as the grasp direction $\mathbf{G_v}$. Motivated by this, we propose a novel view-aware grasp affordances prediction module to predict the in-plane rotation $\mathbf{G}_r$, grasp quality $\mathbf{G}_q$ and gripper width $\mathbf{G}_w$ given a specific grasp center $\mathbf{G_p}$ and grasp direction $\mathbf{G_v}$.

Specifically, given a depth image $\mathbf{D}_t \in \mathbb{R}^{H \times W}$ captured by the depth camera on a robotic arm, we first integrate it into a TSDF $\mathbf{M}_t \in \mathbb{R}^{40 \times 40 \times 40}$, which represents a cubic workspace of size $L$ and can be incrementally updated with the new observed depth images. As shown in Fig. 3, at each time step $t$, our model takes the current TSDF voxel $\mathbf{M}_t$ as input and processes it with a 3D CNN network to obtain a tri-plane feature volume $\mathbf{C} \in \mathbb{R}^{h \times w \times 3}$ as in [32], which is shared for view-aware grasp affordances prediction and novel view depth synthesis.

For grasp affordances prediction of the occluded target object, we first uniformly sample $N$ points in 3D as grasp centers $\mathbf{G_{p_1}}, \mathbf{G_{p_2}}, ..., \mathbf{G_{p_N}}$ in the given 3D bounding box, and set the current observation view $\mathbf{O_v}$ to grasp view $\mathbf{G_v}$. To predict grasp affordance for a specific grasp center $\mathbf{G_p}$, we cast a ray $\mathbf{r} = (\mathbf{G_p}, \mathbf{G_v})$ from orthographic cameras [33] origins $\mathbf{o}$ along the direction $\mathbf{G_v}$ passing through the grasp center $\mathbf{G_p}$. In particular, as shown in Fig. 3, these rays are cast into the tri-plane feature volume $\mathbf{C}$ and $n$ points $\mathbf{S} = \{\mathbf{s}_1, \mathbf{s}_2, ..., \mathbf{s}_n\}$ are uniformly sampled along each ray. Then, we query the local features $\mathbf{C_{s_i}}$ of these 3D points from the shared tri-plane feature volume, and these local features are integrated together as a ray feature $\mathbf{C}_{\text{ray}}$ via max pooling, i.e.,

$$\mathbf{C}_{\text{ray}} = \text{maxpool}(\mathbf{C_{s_1}}, \mathbf{C_{s_2}}, ..., \mathbf{C_{s_n}}). \tag{1}$$

This ray feature captures the occlusion relationships between objects along the ray direction, which is essential for grasp affordances prediction in cluttered scenes.

In addition, local geometry information around the grasp center is another key factor for grasp affordances prediction on a target object. Therefore, as shown in Fig. 3, we draw a small 3D bounding box along the view-direction $\mathbf{O_v}$ with fixed length and width around grasp center $\mathbf{G_p}$. Then we obtain the tri-plane features of the eight vertices of this cuboid $\mathbf{C_{vert_1}}, \mathbf{C_{vert_2}}, ..., \mathbf{C_{vert_8}}$ and concatenate these features with the feature of grasp center $\mathbf{C_{G_p}}$. This concatenated feature is denoted as local geometry feature $\mathbf{C}_{\text{geo}}$, i.e.,

$$\mathbf{C}_{\text{geo}} = \text{concat}(\mathbf{C_{vert_1}}, \mathbf{C_{vert_2}}, ..., \mathbf{C_{vert_8}}, \mathbf{C_{G_p}}). \tag{2}$$

Based on the above ray feature and local geometry feature, we implement the grasp affordances prediction module as a small fully-connected neural network $f_{\mathbf{G}}$ that takes $\mathbf{G_v}, \mathbf{C}_{\text{ray}}, \mathbf{C}_{\text{geo}}$ as input and outputs the in-plane rotation $\mathbf{G}_r$, grasp quality $\mathbf{G}_q$ and gripper width $\mathbf{G}_w$,

$$\mathbf{G}_r, \mathbf{G}_q, \mathbf{G}_w \leftarrow f_{\mathbf{G}}(\mathbf{G_v}, \mathbf{C}_{\text{ray}}, \mathbf{C}_{\text{geo}}). \tag{3}$$

In (3), $\mathbf{G_v}$ is a 3-dimensional unit vector that represents the view direction, $\mathbf{G}_w \in [0, w_{\text{max}}]$ where $w_{\text{max}}$ is the maximum gripper width, and grasp quality $\mathbf{G}_q \in [0, 1]$.

## 4.2 Affordance Imagery with Implicit Neural Representation

Inspired by the impressive performance of neural radiance fields in the new view synthesis, we adopt the same paradigm to enable our model to imagine the scene geometry from previously unobserved viewpoints. With this scene imagery ability, our model can predict reasonable grasp affordances under unseen viewpoints, and the imagined grasp affordances are used for the next-best-view selection. Specifically, as in Fig. 3, we share the same tri-plane feature volume $\mathbf{C}$ for novel view depth synthesis and grasp affordances prediction, and the network is trained with two tasks simultaneously.

First, we build a geometry decoder upon the shared tri-plane feature volume $\mathbf{C}$ for novel view depth synthesis. We implement this geometry decoder as an MLP network that takes local feature $\mathbf{C}_{xyz}$ of a 3D point $\mathbf{p} = (x, y, z)$ as input and predict its signed distance function (SDF) value. Then, for a given view direction $\mathbf{D_v}$, we sample a series of 3D points along the ray and synthesize depth images $\mathbf{D}$ using their corresponding SDF values, following the approach described in NeuS [34], i.e.,

$$\mathbf{D} \leftarrow F_{\mathbf{S}}(\mathbf{C}, \mathbf{D_v}), \tag{4}$$

where $F_{\mathbf{S}}$ denotes the whole network branch for novel view depth synthesis. Note that here we only utilize depth images as supervision, which differs from the approach described in the original NeuS paper. The optimization with (4) makes the shared tri-plane feature volume able to reason scene geometry under unseen viewpoints, which aids us in grasping affordances imagery.

For grasp affordances imagery, we utilize the method described in (3) to predict grasps at points in the bounding box $\mathbf{T}_{\text{bbox}}$ from given direction $\mathbf{G_v}$. The model takes current feature volume $\mathbf{C}$ as input and imagine a grasp affordance map for a novel view. Let $F_{\mathbf{G}}$ represent the whole network branch for grasp affordances prediction, the affordances imagery pipeline is formulated as:

$$\mathbf{G} \leftarrow F_{\mathbf{G}}(\mathbf{T}_{\text{bbox}}, \mathbf{C}, \mathbf{G_v}). \tag{5}$$

## 4.3 Next-Best-View Planning for Grasping

We design a closed-loop next-best-view planning policy $\pi$ to determine the next observation view which is most beneficial for grasping the target object when no feasible grasp is found. Let $\mathbf{G_v}$ be a view from a set of potential next observation views $\mathbb{G_v} \subset SE(3)$. The goal of our next-best-view planning policy is to find the next observation view $\mathbf{O}_{v,t+1}$ with the highest predicted grasp quality $\mathbf{G}_q^*$ from a set of imagined grasp affordances for the target object, i.e.,

$$\mathbf{O}_{v,t+1} \leftarrow \underset{\mathbf{G_v} \in \mathbb{G_v}}{\arg\max} \mathbf{G}_q^*(\mathbf{T}_{\text{bbox}}, \mathbf{C}_t, \mathbf{G_v}). \tag{6}$$

We adopt a methodology similar to that presented in [4] to generate potential next grasping views $\mathbb{G_v}$, and predict the imagined grasp affordances under these potential views with the above affordances imagery module. In addition, we use two simple stopping criteria to decide whether to stop or to continue to find the next observation view. First, the policy is terminated if the highest grasp quality $\mathbf{G}_q^*$ of currently predicted grasps is above a given threshold $q_{\text{max}}$. Second, we impose a maximum number of next-best-view planning steps $T_{\text{max}}$. We summarize the overall next-best-view planning policy for grasping as an algorithm in the appendix.

## 4.4 Training

The network is trained end-to-end using ground-truth grasps obtained through simulated trials. To achieve generalizable grasp affordances imagery, we generate three types of input-output data

pairs: **front-observe-front-grasp**, **front-observe-side-grasp**, and **multi-observe-front-grasp**. The **front-observe-front-grasp** means that our model takes one front view depth image as input and predicts grasp affordances and depth image under the same front view. Similarly, the other two types of data pairs represent different input and prediction task pairs under different views. Note that multi-observe means the input TSDF is fused from several depth images from different views, aiming to construct data that closely resembles the input distribution during the reasoning process of the closed-loop grasping. By incorporating such data pairs into the dataset, we enable the model to predict the affordance of objects in unobserved directions, thus allowing for a more accurate evaluation of candidate observation directions.

The training loss consists of two components: the grasp affordances prediction loss $\mathcal{L}_A$ and the novel view depth synthesis loss $\mathcal{L}_S$. For the grasp affordances prediction loss, we adopt a similar training objective as VGN [1]:

$$\mathcal{L}_A(\mathbf{G}, \hat{\mathbf{G}}) = \mathcal{L}_q(\mathbf{G}_q, \hat{\mathbf{G}}_q) + \mathcal{L}_\mathbf{r}(\mathbf{G}_r, \hat{\mathbf{G}}_\mathbf{r}) + \mathcal{L}_w(\mathbf{G}_w, \hat{\mathbf{G}}_w)). \tag{7}$$

In (7), $\hat{\mathbf{G}}$ represents the ground-truth grasp, and $\mathbf{G}$ represents the predicted grasp. The ground-truth grasp quality is denoted by $\hat{\mathbf{G}}_q$, which takes on a value of 0 (representing failure) or 1 (representing success). The binary cross-entropy loss between the predicted and ground-truth grasp quality is represented by $\mathcal{L}_q$. The cosine similarity between the predicted rotation $\hat{\mathbf{G}}_r$ and the ground-truth rotation $\mathbf{G}_r$ is denoted as $\mathcal{L}_r$, while $\mathcal{L}_w$ represents the $\ell_2$-distance between the predicted gripper width $\hat{\mathbf{G}}_w$ and the ground-truth gripper width $\mathbf{G}_w$. The supervision of the grasp rotation and gripper width is only applied when the grasp is successful (i.e., $\hat{\mathbf{G}}_q = 1$).

The geometry loss is calculated using the standard $\ell_1$ loss between the synthesized depth image and the actual depth image, and is denoted by $\mathcal{L}_S$. The final loss $\mathcal{L}$ is obtained by adding the affordances loss and the geometry loss together, i.e., $\mathcal{L} = \mathcal{L}_A + \mathcal{L}_S$.

## 5  Experiments

We evaluate the performance of our algorithm by grasping an occluded target object in simulation and real-world environments. We use a 7-DoF Panda robotic arm from Franka Emika, with a RealSense D435 attached to the end effector. Our algorithm was implemented in Python, using PyTorch for neural network inference and ROS as the hardware interface. We use Open3D [35] for TSDF fusion, and all experiments use TRAC-IK [36] for IK computations, and MoveIt [37] for motion planning and are run on a same computer.

In our experiments, we evaluate the performance of our method and existing methods with the following metrics. **Success Rate (SR)**: the proportion of successful grasps. **Failure Rate (FR)**: the proportion of failed grasps. **Abort Rate (AR)**: the proportion of cases where no valid grasp was found even after the maximum number of views was reached. **#Views**: the average number of views planned by the algorithm for each round. **Time** (only in real world experiments): the total time consumed, including observation, planning, and execution.

We compare the performance of our algorithm with the following baselines: 1) **initial-view**: most work in visual grasp detection considers a single viewpoint for grasp detection. In this baseline the robot detects a grasp using only the initial view. 2) **top-view**: the robot detects grasps from a single top-down image captured on the top of the workspace center, which is a typical setting for tabletop robotic manipulation. 3) **fixed-traj.**: the robot captures 4 images by moving along a circular trajectory centered on the target object, looking down at 30°, with uniform intervals. The images are subsequently used for TSDF fusion and grasp detection. 4) **GIGA**: a simple next-best-view policy based on GIGA [2], which uses the predicted best grasp direction as the next view direction. 5) **Breyer's**: the state-of-the-art closed-loop next-best-view policy from [4], using the geometry-based information gain approach to plan next-best-view for target object grasping. All baselines use the same controller described above to generate the robot motion.

## 5.1 Simulated Experiments

As shown in Fig. 4, in simulation environments, we generate simulation scenes in PyBullet [38] using the "packed" approach described in [1] and the object with the smallest amount of visible pixels in initial view is selected as the grasping target. Table 1 shows the results of the 400 experiment trials with each method in simulation environments.

We observe that the success rate of the *initial-view* method is the lowest since the strong occlusion of the target object in the initial view leads to a difficult grasp affordance prediction. *Top-view* results in a high success rate because the target can always be grasped from the top in the generated scenes, making it a simple and effective strategy to find feasible grasps. The success rate of the *fixed-traj.* algorithm is higher than the *initial-view* as it collects more scene information from 4 predefined viewpoints. On the other hand, the state-of-the-art closed-loop next-best-view planning method *Breyer's* achieves superior grasping performance and it requires only a few new observations. Finally, compared to *Breyer's*, our method achieves a comparable success rate with fewer new observations and obtains a significant improvement on the success rate when only one new observation (2-Views SR) is allowed. This indicates that our model can find more informative views for grasping the target object. Moreover, the qualitative results of next-best-view planning of our model is shown in Fig. 4, and the examples verify the effectiveness next-best-view planning ability of our proposed method.

To investigate the influence of different components within our model, we test two variants in Table 1: (i) **ours w/o feature $C_{geo}$ and $C_{ray}$** where the features $C_{geo}$ and $C_{ray}$ are replaced with feature $C_{G_P}$ of the grasp center, and (ii) **ours w/o novel view synthesis branch** that removes the novel view depth synthesis in Sec. 4.2. We find that ours w/o features $C_{geo}$ and $C_{ray}$ results in significantly worse grasp affordances prediction, and ours w/o novel view synthesis branch needs more observation views to achieve a comparable performance. Moreover, the results of 2-View SR suggests their importance in finding informative views.

Table 1: Results from the simulation experiments

| Method | SR | FR | AR | #Views | 2-Views SR |
|---|---|---|---|---|---|
| initial-view | 71% | 7% | 22% | 1.00 | N/A |
| top-view | 79% | 6% | 15% | 1.00 | N/A |
| fixed-traj. | 77% | 6% | 17% | 4.00 | N/A |
| Breyer's [4] | 81% | 8% | 11% | 4.31 | 77% |
| Our w/o feature $C_{geo}$ and $C_{ray}$ | 74% | 8% | 18% | 2.54 | 72% |
| Ours w/o novel view synthesis branch | 80% | 10% | 10% | 3.86 | 75% |
| Ours | 83% | 7% | 10% | 2.97 | 80% |

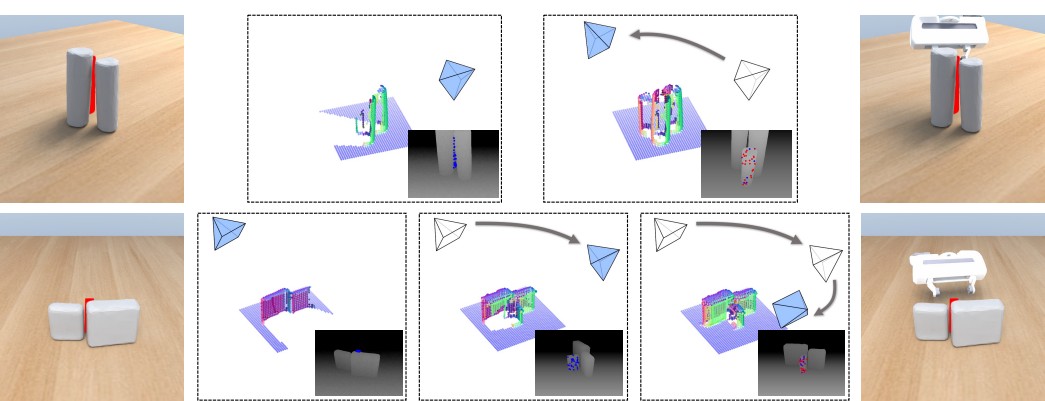

Figure 4: The above images illustrate the next-best-view planning in two simulation scenes. Red and blue pixels in the depth images represent randomly sampled grasp candidates, with red indicating successful grasps and blue indicating unsuccessful grasps as predicted by the model.

## 5.2 Real Robot Experiments

We test our model with a 7-DoF Panda robotic arm in real-world cluttered scenes shown in Fig. 1. We initialize the robotic arm to a position where the target object is partially visible in the initial view. Note that each scene is tested 5 times with small perturbations in the initial robotic arm position and object locations.

The results from 5 grasping trials are reported in Table 2. Intuitively, the difficulty varies among different scenarios. Some objects are heavily occluded, requiring the robot to find suitable directions for observation. Additionally, some objects have unique shapes, limiting stable grasping pose to specific directions. In the relatively easier scenes 4 and 5, our method outperforms the *initial-view* and *fixed-traj.* baselines, while achieving a comparable grasping performance as the state-of-the-art *GIGA* and *Breyer's* method. Furthermore, our algorithm has advantages in much more difficult scenes 1 and 2, where it finds a feasible grasp for the occluded target object with fewer additional observations. For more results, please refer to the supplementary appendix and videos.

Table 2: Real-world experiments setup and results. The first column shows the arrangement of the scene, and the second column displays the view from the initial position of the robotic arm. The target object has been circled with a red dashed line. Our method is capable of achieving comparable grasp success rates (SR) using fewer views (#Views).

| Setup | Initial | Method | SR | FR | AR | #Views | Time/s |
|---|---|---|---|---|---|---|---|
| | | top-down | 0/5 | 3/5 | 2/5 | 1.0 | 18.3 |
| | | fixed-traj. | 3/5 | 2/5 | 0/5 | 4.0 | 29.4 |
| | | GIGA [2] | 3/5 | 1/5 | 1/5 | 5.4 | 32.1 |
| | | Breyer's [4] | 4/5 | 0/5 | 1/5 | 4.8 | 24.7 |
| | | Ours | 3/5 | 2/5 | 0/5 | 3.4 | 23.1 |
| | | top-down | 1/5 | 2/5 | 2/5 | 1.0 | 18.0 |
| | | fixed-traj. | 3/5 | 1/5 | 1/5 | 4.0 | 30.5 |
| | | GIGA [2] | 4/5 | 1/5 | 0/5 | 3.6 | 31.8 |
| | | Breyer's [4] | 3/5 | 0/5 | 2/5 | 5.2 | 25.9 |
| | | Ours | 4/5 | 1/5 | 0/5 | 3.0 | 22.2 |
| | | top-down | 2/5 | 2/5 | 1/5 | 1.0 | 16.5 |
| | | fixed-traj. | 4/5 | 1/5 | 0/5 | 4.0 | 29.8 |
| | | GIGA [2] | 4/5 | 0/5 | 1/5 | 3.8 | 30.2 |
| | | Breyer's [4] | 3/5 | 2/5 | 0/5 | 4.8 | 23.0 |
| | | Ours | 5/5 | 0/5 | 0/5 | 3.2 | 24.7 |
| | | top-down | 0/5 | 0/5 | 5/5 | 1.0 | 19.1 |
| | | fixed-traj. | 2/5 | 3/5 | 0/5 | 4.0 | 30.5 |
| | | GIGA [2] | 3/5 | 1/5 | 1/5 | 5.0 | 28.6 |
| | | Breyer's [4] | 2/5 | 1/5 | 2/5 | 4.4 | 23.2 |
| | | Ours | 3/5 | 2/5 | 0/5 | 2.8 | 23.7 |
| | | top-down | 3/5 | 2/5 | 0/5 | 1.0 | 15.3 |
| | | fixed-traj. | 4/5 | 1/5 | 0/5 | 4.0 | 29.7 |
| | | GIGA [2] | 4/5 | 1/5 | 0/5 | 1.8 | 23.5 |
| | | Breyer's [4] | 4/5 | 1/5 | 0/5 | 3.0 | 21.7 |
| | | Ours | 5/5 | 0/5 | 0/5 | 2.6 | 20.8 |

## 6 Conclusion and Limitations

In this paper, we introduce a next-best-view planning framework that leverages the imagined grasp affordances to plan the robotic arm's new observation views for grasping a target object in occluded environments. This framework is motivated by the idea that the grasp affordances can be well-predicted when the observation direction is aligned with the grasping direction. Through both simulated and real-world experiments, we demonstrate the effectiveness and robustness of our approach compared to previous works.

**Limitations:** Our next-best-view planning framework involves neural network inference and requires the sampling of multiple views, leading to a high computational cost. In addition, the robotic arm motion planning is not considered in our method, and some unsatisfactory grasp executions exist in real robot experiments. In the future, we plan to integrate motion planning into our method to perform more complex tasks in more challenging environments.

**Acknowledgments**

This work is supported by the Shanghai Artificial Intelligence Laboratory, National Key R&D Program of China (2022ZD0160100) and the National Natural Science Foundation of China (62106183 and 62376222).

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

# Appendix

## A   Pseudo Code of the Proposed ACE-NBV Policy

We summarize the overall NBV planning policy for grasping a target object in the following algorithm 1. In the experiment, we set $T_{\max}$ to 8 and $q_{\max}$ to 0.95.

---
**Algorithm 1** Grasp Affordance Prediction and Next-Best-View Planning

---
**Input:** A cluttered scene, an occluded target object given its 3D bounding box $\mathbf{T}_{\text{bbox}}$
**Output:** A feasible grasp $\mathbf{G}$ of the target object
   **for** $t \leq T_{\max}$ **do**
      $\mathbf{M}_t \leftarrow \mathbf{D}_t$                                     ▷ Intergrate depth image into TSDF
      $\mathbf{C}_t \leftarrow$ 3D CNN$(\mathbf{M}_t)$                      ▷ Encode feature
      **if** $\mathbf{G}_q^*(\mathbf{T}_{\text{bbox}}, \mathbf{C}_t, \mathbf{O}_{\mathbf{v},t}) \leq q_{\max}$ **then**
         $\mathbf{O}_{\mathbf{v},t+1} \leftarrow \arg\max_{\mathbf{G}_{\mathbf{v}} \in \mathbb{G}_{\mathbf{v}}} \mathbf{G}_q^*(\mathbf{T}_{\text{bbox}}, \mathbf{C}_t, \mathbf{G}_{\mathbf{v}})$   ▷ Evaluate candidate next views
         Move camera to $\mathbf{O}_{\mathbf{v},t+1}$                ▷ Go to the next-best-view
      **else**
         Execute grasp $\mathbf{G}^*(\mathbf{T}_{\text{bbox}}, \mathbf{C}_t, \mathbf{O}_{\mathbf{v},t})$
         Break
      **end if**
   **end for**

---

## B   Network Architecture and Implementation Details

We adopt the same encoder as in GIGA that takes TSDF $\mathbf{M}_t \in \mathbb{R}^{40 \times 40 \times 40}$ as input and outputs a feature embedding for each voxel with a 3D CNN layer. Then, the tri-plane feature grids is constructed by projecting each input voxel on a canonical feature plane via orthographic projection. Then, three feature planes are processed with a 2D U-Net that consists of a series of down-sampling and up-sampling 2D convolution layers with skip connections. The output is formulated as the shared tri-plane feature volume $\mathbf{C} \in \mathbb{R}^{3 \times 40 \times 40 \times 32}$, where 32 is the dimension of the feature embedding.

Based on the shared tri-plane feature volume, the local feature $\mathbf{C}_{\mathbf{p}}$ of a 3D point $\mathbf{p} = (x, y, z)$ is obtained by projecting it to each feature plane and querying three features $\mathbf{C}_{\mathbf{p}_x}, \mathbf{C}_{\mathbf{p}_y}, \mathbf{C}_{\mathbf{p}_z}$ at the projected locations using bilinear interpolation, and the local feature $\mathbf{C}_{\mathbf{p}}$ is the concatenated feature of these queried features, i.e., $\mathbf{C}_{\mathbf{p}} = \text{concat}(\mathbf{C}_{\mathbf{p}_x}, \mathbf{C}_{\mathbf{p}_y}, \mathbf{C}_{\mathbf{p}_z})$. We implement our grasp affordance prediction network with a five layer fully-connected network with residual connections. The input dimension of this MLP network is $3 + 96 + 9 \times 96 = 963$ which is composed of view direction unit vector $\mathbf{v} \in \mathbb{R}^3$, the ray feature $\mathbf{C}_{\text{ray}} \in \mathbb{R}^{96}$, and the local geometry feature $\mathbf{C}_{\text{geo}} \in \mathbb{R}^{9 \times 96}$. The output dimension for grasp affordance prediction is $1 + 1 + 1 = 3$ which includes the grasp quality $\mathbf{G}_q$, in-plane rotation $\mathbf{G}_r$, and gripper width $\mathbf{G}_w$.

As for the novel view depth synthesis, we employ a MLP network that takes the 3D point feature $\mathbf{C}_{\mathbf{p}} \in \mathbb{R}^{96}$ as input and output the SDF value of this point, and adpot the same rendering technique with NeuS to synthesize depth images ($\eta = 12, \gamma = 5$). We sample 128 rays in a depth image in each batch, each ray consisting of 64 uniformly sampled points and extra $4 \times 32$ points following the importance sampling rule. We set the near and far range close to the ground truth depth at the beginning of training, and then gradually relax the range to the maximum range of the implicit feature volume.

For experiments in simulation and real word, the size of cubic workspace $L = 30$cm. The size of the cubic for $\mathbf{C}_{\textbf{vert}}$ is 0.25, which is 7.5cm in the real world. The points $\mathbf{S} = \{\mathbf{s}_1, \mathbf{s}_2, ..., \mathbf{s}_n\}$ for $\mathbf{C}_{\textbf{ray}}$ is uniformly sampled with a step of 0.1. The sizes of the three datasets front-observe-front-grasp, front-observe-side-grasp and multi-observe-front-grasp are 1M, 1M and 2M grasps, respectively. Each scene contains 240 grasps and $\eta = 12$ ground-truth depth images with the resolution of $480 \times 640$. After data cleaning and balancing, there are about $40\%$ data left. We separate the datasets randomly into 90% training and 10% validation. We train the models with the Adam optimizer and a learning rate of $2 \times 10^{-4}$ and batch sizes of 128. All experiments are run on a computer equipped with an Intel Core i9-13900K and a GeForce RTX 4090.

## C   Extra Experiments for Intuition

We conducted extra experiments in simulation to justify our intuition that the grasp affordances can be better-measured using the observation when the observation view direction is the same as the grasp view. In each randomly selected case, the network receives a depth image from different directions and is then required to predict a grasping pose of the target object in the same direction. The results are quite evident: the prediction is much better when those two directions are the same.

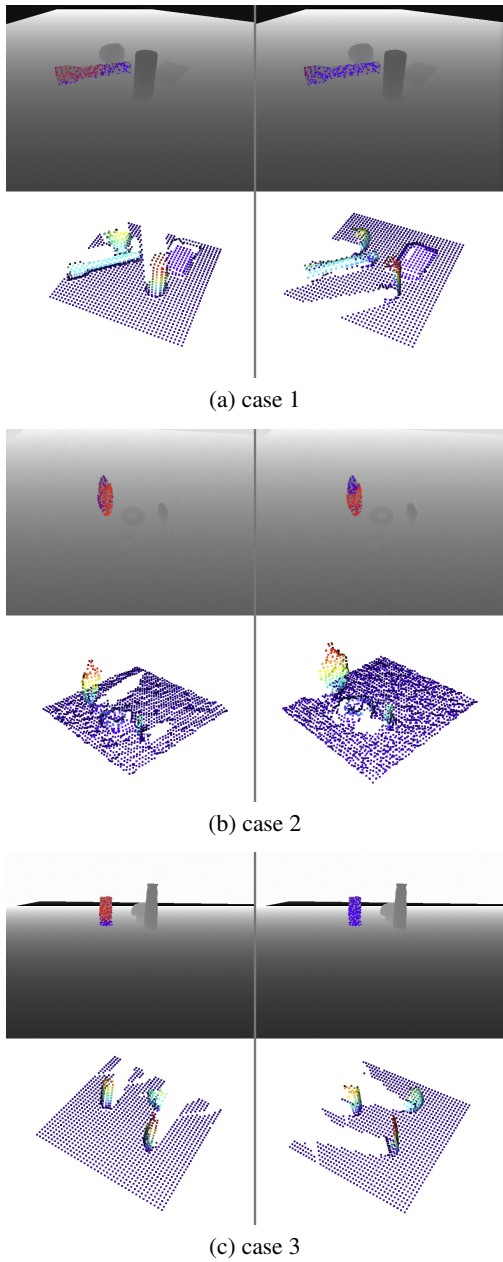

(a) case 1

(b) case 2

(c) case 3

Figure 5: The network receives a TSDF fused from a depth image captured either from the front (lower left) or the side (lower right) as input. It is required to predict grasp in the frontal direction, which aligns with the direction of the depth image used for visualization in the upper row. The color red indicates high-quality grasps, while blue represents low-quality ones.

# D Qualitative Results of Real Robot Experiments

We present qualitative results in Fig. 6 and 7 and recommend readers watch the supplementary video for more comprehensive real robot experimental results. Note that our model can select reasonable next-best-view to observe the occluded target object. We show a representative failure case in Fig. 7, where small errors in grasp affordance prediction leads to an unsuccessful grasp. This small prediction inaccuracy occurs in most failure experiments. Therefore, in the future, we plan to exploit a better grasp affordance prediction module to improve the success rate of our method.

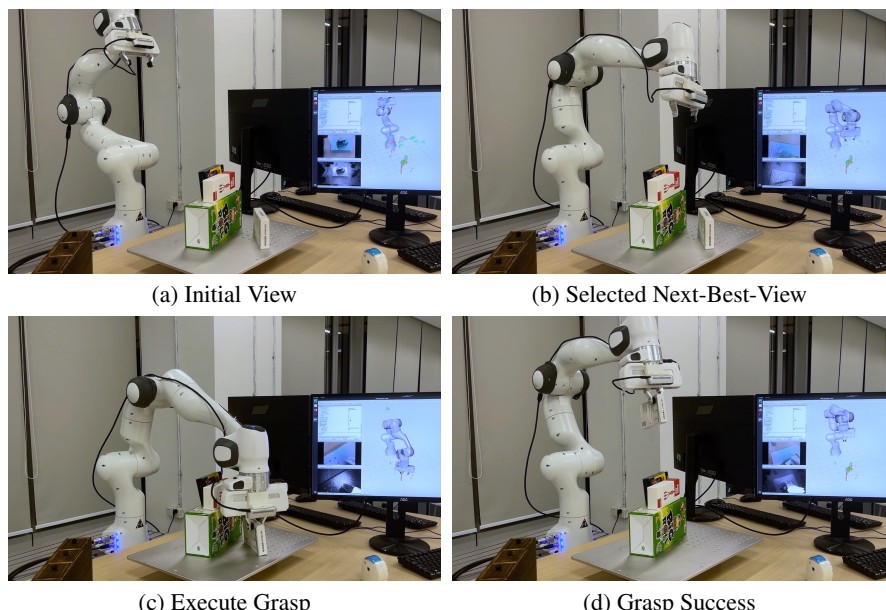

(a) Initial View          (b) Selected Next-Best-View

(c) Execute Grasp          (d) Grasp Success

Figure 6: Success Case. The robot planned one new view to observe the target box and successfully grasped it.

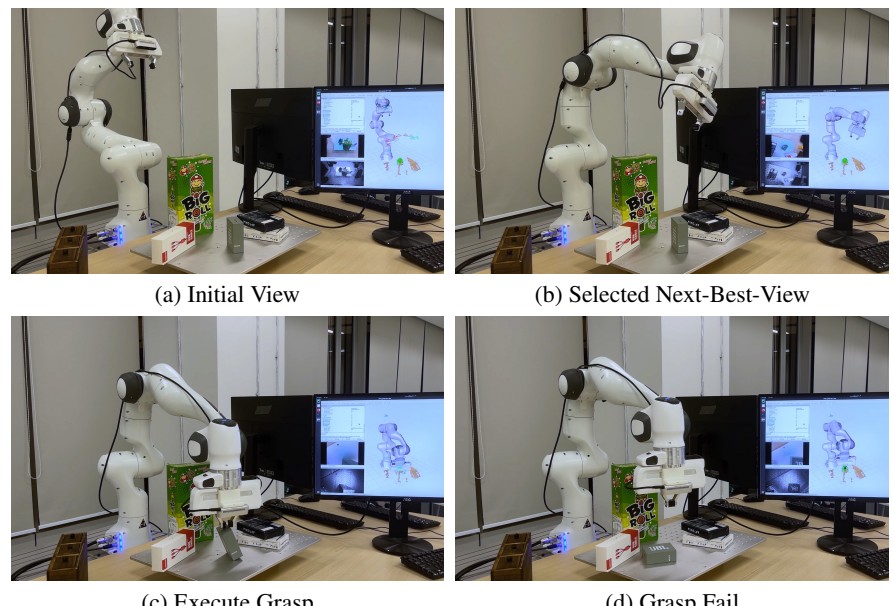

(a) Initial View          (b) Selected Next-Best-View

(c) Execute Grasp          (d) Grasp Fail

Figure 7: Failure Case. The robot failed to predict accurate grasp affordances of the target object after obtaining a new observation. As a result, the grasping failed.

