# OpenReview forum: "Affordance-Driven Next-Best-View Planning for Robotic Grasping"
_robot-learning.org/CoRL/2023/Conference — CoRL 2023 Poster_

### Official Review · Reviewer_3RPC · 2023-07-13

**Confidence:** 4
**Originality:** Good
**Technical Quality:** Good
**Clarity Of Presentation:** Good
**Impact:** 3

**Recommendation:**

Weak Accept: I recommend accepting the paper, but will not argue for my recommendation if the majority of other reviewers have a different opinion.

**Review:**

While grasping is not a new task, and in fact has been one of the most studied robotic tasks, combining it with a NeRF-based next-best-view method is. Grasping in cluttered scenes is hard, and this paper proposes a novel way to solve that task. By utilizing the underlying 3D representation to predict where the next best grasp might be, the robot is able to move the camera to that location to better estimate the grasp quality. This is an interesting idea. Furthermore, a major strength of this paper is the extensive use of baselines including ablations of the proposed methodology as well as a completely different method from prior work. Results are shown across all methods and for both sim and real environments. They make a convincing argument as to why the proposed method is superior to alternatives.

The biggest weakness of the paper is that a few of the components are relatively simplistic. The grasp policy assumes the gripper travels along a straight line, closes the gripper, and presumably travels directly back. While this might work for some cluttered scenes, it won’t work in many more, e.g., a cluttered kitchen cabinet. Additionally, the next-best-view planner utilizes a greedy approach, i.e., it picks the view that predicts the highest likelihood of a successful grasp. Why is this the best way to pick the next view? Why not use another metric like information gain? Or maybe even a planner that predicts several views? Furthermore, the results only evaluate on 4 fairly similar scenes on the real robot (detailed more in the rebuttal section). To be clear, none of these are deal-breakers, merely observations.


**Quality Of The Limitations Section:**

Limitations are addressed clearly

**Questions For Rebuttal:**

For the rebuttal, the authors should make the following changes:

Section 4.2 is quite unclear. How are depth images generated by sampling SDF values at individual points in the tri-plane? Why/how are the number of images downsampled from \eta to \gamma and then what exactly is F_s doing to return to \eta number of images? Also, the observe-grasp data pair described in the last paragraph don’t seem to be related to the rest of the section. Is this a data collection detail? Should this paragraph belong in section 4.4? I would strongly recommend having a colleague who is not already familiar with this work proofread this section and then rewrite it for clarity.

There are only 4 real-world scenes, and they’re all pretty similar: the target object is hidden behind one or more large objects. There really should be more scenes with more variation. For a sanity check, this should be tested on a scene with the target object in front of other objects just to make sure it works in simpler cases. It’s also worth testing having the target object completely surrounded on all sides, requiring the robot to reach down from above. Additionally, other tests could include creating a “U” shape of objects around the target, forcing the robot to pick from either above or the opening of the “U”. There are many other potential variations that could be run and I leave it up to the authors to determine them, but I strongly recommend adding at least 4 scenes, if not more, with significant variation.

Finally, this last issue is more of a pedantic one: This paper doesn’t use NeRFs (neural radiance fields), key word here being “radiance”. The methodology here, as far as I can tell, exclusively uses depth images, but radiance is never computed or used. The correct term for what this paper uses would be Neural SDF. Please fix the terminology.


**Robotics Focus:**

Highly relevant to robotics but no hardware experiments

**Summary Of Paper:**

This paper proposes a method for using novel view synthesis to plan the best grasp of an object in a cluttered scene. Given a 3D bounding box of the target object in the cluttered scene, first a TSDF is built from the RGBD observation. Then a learned grasp success model is queried. If it finds a grasp it predicts will be successful, the robot executes the grasp and the trial is over. If not, the learned Next-Best-View model is queried to predict the next view to move the camera to. The RGBD image is integrated into the TSDF and the loop is repeated. The results show that the proposed method outperforms the baseline in either success rate or number of required views or both.


**Summary Of Recommendation:**

This paper presents an interesting and novel methodology, and does an excellent job comparing to several baselines. If the authors are able to address my comments in the rebuttal section, my recommendation would be weak accept. The clarity of section 4.2 and the number of real world scenes are the biggest issues to address.

---

### Official Review · Reviewer_pq4R · 2023-07-18

**Confidence:** 5
**Originality:** Good
**Technical Quality:** Good
**Clarity Of Presentation:** Good
**Impact:** 3

**Recommendation:**

Weak Accept: I recommend accepting the paper, but will not argue for my recommendation if the majority of other reviewers have a different opinion.

**Review:**

Strengths:

- The extension of GIGA [2] to a view-dependant grasp affordance function is interesting to the community
- Apart from the methodology section, the paper is well written
- The figures in the paper are useful for understanding the approach
- The authors have carried out real-robot experiments with a Franka robot with a cluttered table-top scene
- The paper clearly mentions some limitations

Weaknesses:

- The methodology section is not so well-written, making the overall approach difficult to understand. For example, the authors should explain clearly that the network is jointly trained for reconstruction (depth synthesis) and grasp affordance prediction to leverage synergies between these two objectives (as shown in [2]). Moreover, it should be made clear in section 4.2 that the depth samples are taken during the data-generation phase.
- There are issues with the Affordance Imagery part of the methodology:
    - First, the authors should refrain from using the term Neural “Radiance” since the approach does not use radiance and only geometry or surface.
    - Secondly, the authors mention the rendering approach NeuS by Wang et al. [28], but the NeuS approach was mainly meant for 2D RGB images without any 3D supervision. The authors should clarify that they use the renderer with depth images instead.
    - Thirdly, the authors should justify why they opted to use a Volume Rendering technique instead of a Surface Rendering technique, which is typically faster and better suited to depth images.
    - It is unclear why the authors use depth-image supervision to train the SDF instead of directly using ground-truth SDF values from the simulation. Using ground truth SDF values OR occupancy values (as in GIGA [2]) would directly solve the problem of learning appropriate geometric features in the scene. It would also be more computationally efficient since the expensive volume rendering step would be avoided.
- While a comparison has been made with the method from Breyer et al. [1], a comparison with GIGA [2] is missing. Testing the method against GIGA is important since it also uses the scene reconstruction head. A simple next-best-view method using GIGA can be devised by just using a view that “looks at” GIGA’s predicted best grasp i.e. view direction is the same as grasp view orientation G_v and centered at the grasp’s position center.
- Missing citations:
    - Recent related grasping methods:

    Huang, H., Wang, D., Zhu, X., Walters, R., & Platt, R. (2023, May). Edge grasp network: A graph-based se (3)-invariant approach to grasp detection. ICRA 2023

    Jauhri, S., Lunawat, I., & Chalvatzaki, G. (2023). Learning Any-View 6DoF Robotic Grasping in Cluttered Scenes via Neural Surface Rendering. arXiv preprint arXiv:2306.07392.

    Dai, Qiyu, et al. "GraspNeRF: multiview-based 6-DoF grasp detection for transparent and specular objects using generalizable NeRF." ICRA 2023
    - Given that a tri-plane feature volume is used, also cite Peng et al.

    Peng, Songyou, et al. "Convolutional occupancy networks." Computer Vision–ECCV 2020: 16th European Conference, Glasgow, UK, August 23–28, 2020, Proceedings, Part III 16. Springer International Publishing, 2020.
- Typo: “Neural radiation fields”

**Quality Of The Limitations Section:**

Limitations are addressed clearly

**Questions For Rebuttal:**

- How well does the method perform when directly trained with ground-truth SDF values for the SDF reconstruction?

- How expensive is the NeuS volume rendering step?

- How well does the method perform compared to a simple GIGA[2]-based next-best-view setup?
i.e. 1. Predict best grasp with GIGA -> 2. Move the franka gripper (and camera) to a view direction that is the same as the predicted best grasp’s orientation -> 3. Execute the grasp if it has high-quality else continue


**Robotics Focus:**

Sufficient demonstration on hardware

**Summary Of Paper:**

In this paper, the authors propose an active perception pipeline for grasping in a cluttered scene. Building on the work of Breyer et al. [1] and Jiang et al. [2], the authors propose using a view-dependant grasp affordance function to detect grasps and use an SDF-based reconstruction head to encourage reconstruction for the prediction of grasp affordance from unseen views. The proposed approach ACE-NBV compares favorably to the work of Breyer et al. [1] and some other simple next-best-view strategies.

**Summary Of Recommendation:**

While the authors propose some interesting ideas, there are some odd choices used in the methodology of novel-view depth prediction that have not been justified in the paper. Moreover, a comparison with a significant baseline is missing due to which the paper cannot already be accepted in its current form.

---

### Official Review · Reviewer_hPBU · 2023-07-19

**Confidence:** 3
**Originality:** Fair
**Technical Quality:** Fair
**Clarity Of Presentation:** Fair
**Impact:** 3

**Recommendation:**

Weak Reject: I recommend rejecting the paper, but will not argue for my recommendation if the majority of other reviewers have a different opinion.

**Review:**

Strength

- The idea of using NeRF for NBV planning is interesting
- Related work is sufficiently addressed
- Real-world experiments are conducted and limitations are discussed

Weakness

- The evaluation cases are overly simplified, and limited to objects with basic geometry (box and cylinder) in the scene. The author may consider incorporating more complex objects (e.g. toy) similar to those used in Breyer's paper.
- Given the simplicity of the evaluation scenarios, a top-view or a randomly sampled fixed 2 or 3 near-top-view might be sufficient to predict the grasping pose. However only fixed 4-view are discussed (in both simulation and real-world experiments), top-view is not discussed in real-world experiments. Moreover, The author may consider including more details on how the fixed views are generated. Well-generated fixed views, such as those randomly generated around top-view, have the potential to perform effectively across different scenarios. In contrast, poorly generated fixed views, like multiple side-views from the same direction and close to each other, may work poorly.
- The use of bounding box of the occluded objects seems to be not realistic - if the target object is fully (or mostly) occluded, the bounding box cannot be detected from the current observation. However, information-based NBV method could still work in these scenarios by simply finding the next view that maximizes the information, until observing the occluded object. The author may consider evaluation scenarios with fully occluded target objects.
- The proposed method is not showing significant advantages over baselines given the current result in simple scenarios, and considering the baseline's effectiveness in fully occluded scenarios. The author may consider conducting further experiments in more complex scenarios to demonstrate the superiority of the proposed method.
- The author claims "it is intuition that the grasp affordances can be better-measured using the observation when the observation view direction is the same as the grasp view". However, it doesn't seem to be intuitive, as human rarely use observation view same as grasp view (e.g. grasping a bottle). The author may consider conducting experiments to justify this, for example, using two affordance networks with shared weights on local features to separately predict the grasping pose and next-best-view pose.


**Quality Of The Limitations Section:**

Additional details required

**Questions For Rebuttal:**

- In Eq.1, why the ray feature captures the occlusion relationships? A feature that captures occlusion relationships should be sensitive to the order of input points. However, swapping the order of two neighboring points (S_i, S_{i+1}) may not affect the maxpool results, but the occlusion relationship between S_i and S_{i+1} is changed after swapping.
- Please provide more details explaining what is front-observe / side-observe in Eq.5 and why using them.
- What's the definition of Gq*? It appears to be inconsistent in the paper: Gq*(Tbbox,Ct,Ov,t) and Gq*(Tbbox,Gv) (Algorithm 1 in Supplementary)

**Robotics Focus:**

Sufficient demonstration on hardware

**Summary Of Paper:**

This paper introduces ACE-NBV, a method for next-best-view (NBV) planning in cluttered environment. The proposed architecture is similar to GIGA, with the exception of using affordance module to jointly predict and grasping pose and the next-best-view pose. Specifically, the model integrate depth images into TSDF and then converted into a tri-plane feature volume using a 3D CNN encoder. Novel view depth synthesis following NeRF paradigm is used to aid in learning the encoder. Grasp affordance and grasping params (rotation, gripper width) are predicted given the learned feature volume, along with randomly sampled grasp candidates within the given object bounding boxes of the target object. When the proposed grasp fails to meet the criteria, the next camera poses are generated using the proposed grasp pose.

**Summary Of Recommendation:**

Based on the strengths and weaknesses section, I would currently recommend weak rejection for the paper, despite the idea of using NeRF for NBV planning is interesting. More experiments are needed to demonstrate the advantages of the proposed method.

---

### Official Review · Reviewer_6CeP · 2023-07-20

**Confidence:** 4
**Originality:** Fair
**Technical Quality:** Fair
**Clarity Of Presentation:** Very Good
**Impact:** 3

**Recommendation:**

Weak Accept: I recommend accepting the paper, but will not argue for my recommendation if the majority of other reviewers have a different opinion.

**Review:**

Strengths
- The idea of using grasp affordances from imagined views to guide visual exploration of scene before executing a grasp is a good idea
- The figures in this paper were easy to follow

Weaknesses
- An important missing citation: https://arxiv.org/abs/1912.04344, this paper from 2020 takes a very similar approach to the problem by rendering a TSDF from different views and predicting grasp affordances from those novel views, with which a closed-loop grasp is made. I feel that the wording in this submission that "we, for the first time, introduce an affordance-driven NBV planning policy" is inaccurate. The idea of using grasp affordances from novel views is precisely the idea this prior work proposes.
- I'm confused why the authors refer to their method as using a Neural Radiance Field. The 3D representation used is a TSDF, and the feature volume output by the CNN encoder is not a NeRF, but rather a feature volume from which grasp properties and SDF values are rendered. Rendering this volume from different views does not make it a NeRF, as there is no  radiance represented anywhere in the pipeline.
- It seems to me that taking in a 3D bounding box as input breaks the assumption that the object is occluded, as constructing such a bounding box would require full knowledge of the extents of the object, defeating the purpose of the occlusion-tolerant capabilities. This should be discussed in the limitations section, how could such a box be obtained automatically without observing the occluded part of the object? Could the system be adapted to work with other methods of specifying a target object?
- The experimental evaluation in physical settings is lacking, since only 4 scenes are considered (each scene is repeated 10 times, but it's unclear how significant the "small perturbations" in arm and object positions have on the experiment diversity). In addition, the method provides fairly marginal success improvements over baselines, even over a single-view top-down baseline.

Page 2 line 84: "neural radiation fields"

**Quality Of The Limitations Section:**

Additional details required

**Questions For Rebuttal:**

- The writing should clearly describe the difference in approach between this work and the "grasping in the wild" paper provided above.
- In the videos it seems like many times the first view selected is a top-down view, after which a grasp is carried out. Indeed, in the simulated results the difference between top-view and this method is only 4%, which is hardly significant. The difference between 2-view and top-view is even less (1%). In addition, top-view is not provided in the physical experimental results. Does this method actually present an improvement over a single top view in the physical experiments provided?
- What is Tmax? Providing it would be helpful in contextualizing the number of views taken during experiments.
- Additional experiments with more cluttered scenes where top-down grasping fails would more convincingly show the benefit of this method.
- Including the time taken during experiments would be a useful metric to compare, since lower number of actions doesn't necessarily always mean faster.

**Robotics Focus:**

Sufficient demonstration on hardware

**Summary Of Paper:**

This paper studies closed-loop view selection for grasp planning, where the goal is to grasp a target object which might be partially occluded initially. To address this, they use a continually updating TSDF scene representation, which is converted to a feature field via a 3D CNN. To compute a new view, this feature volume is rendered from multiple viewpoints and grasps are sampled with the grasp axis facing these viewpoints. If grasp confidence is sufficiently high, a grasp is executed, otherwise the next highest confidence view is used as a new viewpoint.

**Summary Of Recommendation:**

The approach is a reasonable strategy for affordance-based view selection, however it is not, as the authors claim, "the first" to propose affordance-driven next-best-view policy. I have some additional concerns with the marginal benefits shown in experiments compared to single view top-down planning in simulation (which is missing in real). If top-view planning is successful in all these scenes, it's difficult to justify the added complexity of this system over a heuristic top-down approach. In that case, more test scenes should be evaluated on to illustrate the benefits of multi-view planning. I also feel the choice of describing this method as a NeRF is erroneous, since in fact the method doesn't represent radiance anywhere, or reconstruct the scene from multi-view RGB images, but rather uses TSDF fusion from a depth camera.

---

### Author Response · Authors · 2023-08-14
**General Response**

We thank all of the reviewers for their time and insightful comments. Furthermore, we are very glad to find that reviewers generally recognized our idea and clear presentation of our paper.

Meanwhile, we thank all the reviewers for their helpful and constructive feedback to improve the quality of our work.

- Regarding the issue mentioned most frequently in the review comments about needing more experiments [6CeP, hPBU, 3RPC], we have provided additional supplementary experiments. Among them, some scenarios fully demonstrate the superiority of our algorithm.
  - [3RPC] We also followed the suggestion from reviewer 3RPC and constructed some complex scenarios where the top-down view fails to provide effective grasping.
  - [6CeP] We have added time metric in the extra experiments.
  - [pq4R] We have added a comparison with simple next-best-view policy using GIGA.

- For the concerns about the writing, we have incorporated the detailed suggestions from all the reviewers and made respective modifications. Specifically, it includes:
  - [6CeP, pq4R] We have added all the important citations and provided clear descriptions accordingly.

  - [6CeP, pq4R, 3RPC] We have corrected the misused terminology “radiance” and changed the keyword to Neural SDF.

  - [6CeP, hPBU] We have added the missing parameter and the detailed definition of baseline in the experiment section.

  - [hPBU] We have conducted extra experiments to justify our intuition.

  - [hPBU, pq4R, 3RPC] We have rewritten the data generation and depth image rendering part in section 4.2 and section 4.4.

  - [pq4R] We have added the relationship and differences between our method and NeuS in section 4.2.

Besides, we would carefully update our paper in the next version to incorporate the extra experiments and related descriptions.

- The original "Real Robot Experiment" in Section 5.2 will be replaced by ExtraExp 1. The original experiment will be included in the appendix to showcase additional scenarios.

- ExtraExp 2 for intuition, following the suggestion from Reviewer hPBU, will be added to the appendix to enhance our idea.

We hope to have addressed all the raised concerns and would be happy to respond to further questions and suggestions.

---

> ### Author Response · Authors · 2023-08-15
> **New version of the article has been uploaded**
>
> The new version of the paper has been uploaded now, which includes additional experiments. We kindly request the reviewers to check for the updated document. We believe that the inclusion of these new experiments has significantly enhanced the quality of our paper.
>
> Once again, we sincerely appreciate the reviewers' valuable input and constructive criticism.

---

### Decision · Program_Chairs · 2023-08-30

**Decision:**

Accept (Poster)

**Comment:**

The paper proposes a grasp planner for initially partially occluded object. It continually updates a scene representation and replans grasps based on updates.

Reviewers appreciated the premise of using imagined views to guide exploration, using NeRF for next-best-view planning, clean figures, some real-world evaluation, clear writing (with a few exceptions), and evaluation baselines.

Reviewers raised concerns about missing related work, some method details (such as the bounding volume), simplified/lacking evaluation, and an unclear method section.

The reviewers appreciated the improvements made by the rebuttal, particularly the additional experiments, causing several to raise their assessment, resulting in an accept. The authors are strongly encouraged to ensure all updates make it to the final camera-ready version--this will likely involve making room for the additional experiments in the main content of the paper (and not leaving it to appendices).